# Does Myocardial Atrophy Represent Anti-Arrhythmic Phenotype?

**DOI:** 10.3390/biomedicines10112819

**Published:** 2022-11-04

**Authors:** Barbara Szeiffova Bacova, Katarina Andelova, Matus Sykora, Tamara Egan Benova, Miroslav Barancik, Lin Hai Kurahara, Narcis Tribulova

**Affiliations:** 1Centre of Experimental Medicine, Slovak Academy of Sciences, 84104 Bratislava, Slovakia; 2Department of Cardiovascular Physiology, Faculty of Medicine, Kagawa University, Miki-cho 761-0793, Japan

**Keywords:** cardiac unloading, myocardial atrophy, depressed contractility, arrhythmias, connexin43

## Abstract

This review focuses on cardiac atrophy resulting from mechanical or metabolic unloading due to various conditions, describing some mechanisms and discussing possible strategies or interventions to prevent, attenuate or reverse myocardial atrophy. An improved awareness of these conditions and an increased focus on the identification of mechanisms and therapeutic targets may facilitate the development of the effective treatment or reversion for cardiac atrophy. It appears that a decrement in the left ventricular mass itself may be the central component in cardiac deconditioning, which avoids the occurrence of life-threatening arrhythmias. The depressed myocardial contractility of atrophied myocardium along with the upregulation of electrical coupling protein, connexin43, the maintenance of its topology, and enhanced PKCε signalling may be involved in the anti-arrhythmic phenotype. Meanwhile, persistent myocardial atrophy accompanied by oxidative stress and inflammation, as well as extracellular matrix fibrosis, may lead to severe cardiac dysfunction, and heart failure. Data in the literature suggest that the prevention of heart failure via the attenuation or reversion of myocardial atrophy is possible, although this requires further research.

## 1. Introduction

The heart is an electro-mechanical pump that depends on sinoatrial node pacemaking activity, which initiates the electrical excitation of cardiomyocytes followed by contraction. Changes in mechanical load regulate long-term cardiac function, while the contractile work of cardiomyocytes largely determines cardiomyocyte size. The latter is extremely dependent upon mechanical loading, i.e., stretch and/or tension [1]. In reaction to the circulatory load conditions, myocardial muscle progresses to atrophy or hypertrophy [2]. The structural remodelling of cardiomyocytes is a major process in both cardiac hypertrophy and atrophy in response to increased or decreased workloads [3,4,5]. Thus, the intrinsic plasticity of the cardiac muscle predisposes to compensatory and/or adaptive structural remodelling in response to variations in hemodynamic or mechanical load.

It is generally accepted that the hypertrophy of adult cardiomyocytes results from the adaptation of the heart to pathophysiological conditions. Meanwhile, during the postnatal development of the heart, the small immature cardiomyocytes become larger up to adult cardiomyocyte size in response to alterations in physiological hemodynamic load. Postnatal growth of the human ventricular myocardium is associated with alterations in the spatiotemporal patterns of the cardiomyocyte junctions responsible for electrical coupling (via connexin43 channels) and mechanical coupling (via adhesive junctions) [6]. As such, these junctions play an important role in the electro-mechanical function of the heart.

Enlarged human cardiac chamber size and eccentric or concentric left ventricular hypertrophy is observed in isometric sports activities along with electrocardiographic anomalies [7,8]. In this context, a question arises as to whether the structural remodelling of “athlete’s heart” caused by endurance sports should be considered physiological or pathological hypertrophy? It appears that distinguishing between athlete’s heart and a hypertensive heart can be challenging [9], while hypertension is more prevalent in athletes [10]. Nevertheless, due to genetic background (cardiomyopathy), “athletes heart” is susceptible to developing ventricular arrhythmias, which are mostly benign premature beats due to repolarization (ion channels) abnormalities [11]. Moreover, the occurrence of malignant arrhythmias, such as ventricular fibrillation (VF) or polymorphic ventricular tachycardia, are promoted by structural remodelling [11] with a potential risk for sudden cardiac death [7,12,13]. Thus, it is believed that there is a risk of cardiac arrhythmias regardless of the “physiological” or “pathological” aetiology of myocardial structural remodelling.

Different from myocardial hypertrophy, the molecular mechanisms of cardiac atrophy and their consequences are much less explored. The ins and outs of atrophic remodelling have been previously reviewed [4], focusing mostly on metabolism in the context of a failing heart. However, cardiomyocyte size is likely to be involved in cardiac arrhythmogenesis [14] and the development of malignant arrhythmias. Indeed, experimental and clinical studies suggest that myocardial hypertrophy due to hypertension or hyperthyroidism increases the susceptibility of the heart to life-threatening ventricular tachycardia or fibrillation [15,16,17], while myocardial atrophy due to type-1 diabetes mellitus or hypothyroidism renders the heart less prone to developing malignant ventricular arrhythmias [17,18,19,20]. Thus, it is important to identify the abnormal substrate as a major culprit for sustained ventricular arrhythmias jeopardizing cardiac function.

The intention of this review is to provide state-of-the-art information focusing on the factors and molecular mechanisms underlying myocardial atrophy and their possible relation to the anti-arrhythmic or pro-arrhythmic cardiac phenotypes. Furthermore, tentative strategies to limit or reverse myocardial atrophy and prevent heart failure are outlined.

## 2. Factors and Mechanisms Involved in the Development of Malignant Cardiac Arrhythmias 

In the context of the pro-arrhythmic potential of myocardial hypertrophy, we briefly outlined the mechanisms and key factors involved in arrhythmogenesis that promote the development of life-threatening arrhythmias, such as VF, which causes hemodynamic collapse. 

The fundamental electrophysiological mechanisms of cardiac arrhythmias include abnormal electrical impulse generation, i.e., increased automaticity or triggered activity, and abnormal electrical impulse propagation, i.e., block of conduction and re-entry, whereby these abnormalities coexist [21]. As previously reviewed [21,22], there are three main factors involved in the development of malignant cardiac arrhythmias: arrhythmogenic substrates, triggers and modulating elements. Arrhythmogenic substrates—myocardial structural remodelling, hypertrophy and/or fibrosis along with abnormal topology and impairment of Cx43 channels—promote occurrences of VF due to the slowing or blocking of conduction facilitating re-entrant excitations. Abnormal Ca^2+^ handling, intracellular Ca^2+^ overload, the uncoupling of Cx43 channels and abnormalities of ion currents underlie triggered or ectopic pacemaker-like activity, which initiates VF in the presence of an arrhythmogenic substrate. Meanwhile, sympathetic–vagal imbalance, humoral misbalance (e.g., RAAS), inflammation and redox dysregulation are considered to be modulating elements. Moreover, epigenetic factors such as microRNAs may play a role in the susceptibility of the heart to cardiac arrhythmias [23]. These factors are most likely involved in “physiological” or “pathological” myocardial hypertrophy, thereby affecting the susceptibility of the heart to malignant arrhythmias. 

## 3. Conditions Leading to Myocardial Atrophy and Incidence of Cardiac Arrhythmias

Volume overload is generally associated with eccentric hypertrophy, and increased cardiomyocyte length is the predominant feature. Pressure overload generally results in concentric hypertrophy characterized by an increased cardiomyocyte cross-sectional area. A combined volume and pressure overload can be produced by administration of a thyroid hormone, with an increased length and cross-sectional area of cardiomyocytes [2,24]. Chronic overload of the ventricle results in heart failure and the increased propensity of the heart to life-threatening arrhythmias [15].

Conversely, the structural remodelling of cardiomyocytes leading to decreased cardiomyocyte cross-sectional area and atrophy can be elicited by a reduction in afterload or in a mechanically or metabolically unloaded heart [4,25]. This occurs in thyroid hormone deficiency [19,24,26,27,28,29,30,31,32] or type-1 diabetes mellitus [18,33,34,35] due to an increase in deiodinase-3 that converts both thyroxine and active triiodothyronine into inactive metabolites [36]. Interestingly, atrophic myocardial remodelling due to hypothyroid state and type-1 diabetes is associated with a reduced propensity of the heart to malignant ventricular arrhythmias [17,18,19,20]. Hyperglycaemia ‘per se’ may enhance some pro-arrhythmic mechanisms [37] and increase the risk of sustained atrial arrhythmias [24,38,39], which are related to an enlargement in cardiomyocytes and interstitial fibrosis [40]. However, atrophic cardiomyocyte signalling has been suggested, even in hypertensive heart disease [41]. It occurred at the same time as hypertrophy in the presence of a reparative fibrosis and induction of oxidative and endoplasmic reticulum stress at sites of scarring where myocytes are atrophied.

Moreover, cardiovascular unloading produces cardiopulmonary deconditioning, which may be associated with cardiac atrophy related to spaceflight [42], microgravity conditions [43,44], weightlessness or head-down bed rest [45], while the left ventricle muscle atrophy persists well into recovery. “The sedentary heart”, i.e., physical inactivity, is also associated with cardiac atrophy [46], in contrast to hypertrophy in trained athletes. A left ventricular mass reduction was reported following spinal cord injury [47,48,49] as well as in Alzheimer’s disease models induced in mice [50]. Experimental chronic alcohol exposure [51] and sepsis [52] also induced cardiac atrophy. Moreover, ischemic stroke in mice significantly decreased heart weight and cross-sectional areas of cardiomyocytes and increased atrogin-1 and the E3 ubiquitin ligase MuRF-1, indicating myocardial atrophy [53]. It is not known whether these conditions, with the exception of sepsis, affect the vulnerability of the heart to arrhythmias. However, myocardial inflammation may increase pro-arrhythmic factors [54].

Notably, the negative consequences of the prolonged duration of mechanical support from a left ventricular assist device resulted in myocardial atrophy in patients exhibiting heart failure or non-ischemic cardiomyopathy [55,56]. Myocardial atrophy, which impairs cardiac function, may be one factor limiting the recovery from heart failure in the clinic. In this context, ventricular mechanical unloading was examined in animal models [56,57,58,59], mostly in heterotopic rat heart transplantation. Interestingly, the genetic responses of failing human and healthy rat myocardium to mechanical unloading show similarities [60]. Cardiac arrhythmias are yet to be fully examined.

Left ventricular atrophy may be induced by caloric restriction [61,62] in contrast to hypertrophy associated with obesity. Hypocaloric feeding depresses heart function, causing cardiac atrophy, bradycardia, and decreased cardiac output [63]. Thus, changes in body weight in relation to food intake affect cardiac phenotypes. Prolonged fasting profoundly inhibits the synthesis of new cardiac proteins, including those of constituent myofibrils [64]. Myocardial atrophy, along with oxidative stress due to the intake of saturated fatty acid in mice, has been reported [65] and was suggested as a model of cardiac cachexia.

Atrophic remodelling (“cardiac cachexia”) also occurs during periods of catabolism referring to cancer cachexia [66,67,68,69,70,71,72]. Cardiac atrophy is a hallmark of patients suffering from cancer or some of its treatments, such as anthracyclines, which include doxorubicin. Indeed, the chemotherapy of breast cancer, lymphoma, leukaemia, and sarcoma by anthracyclines reduces left ventricular mass in patients [73,74,75,76,77]. Doxorubicin, when administered to healthy laboratory animals per se reduces heart weight in males [78,79,80,81,82,83,84,85,86,87,88] but not in females [89]. In addition, doxorubicin has a dose-dependent pro-arrhythmic potential mostly due to oxidative stress [90]. However, despite the atrophic remodelling of patients suffering from cachexia or cancer, the incidence of malignant ventricular arrhythmias was not reported. 

Conditions leading to cardiac atrophy and the incidence of cardiac arrhythmias are summarized in Table 1. Altogether, data from the literature suggest that the decrement in left ventricular mass may be the central component in cardiovascular deconditioning and the prevention of life-threatening arrhythmias. Whether ischemic components may impact the process of cardiomyocyte atrophy has not been elucidated. Meanwhile, long-term myocardial atrophy accompanied by oxidative stress and inflammation, along with extracellular matrix fibrosis, leads to cardiac mechanical dysfunction, heart failure progression rather than malignant ventricular arrhythmias.

## 4. Characteristic and Mechanisms Underlying Atrophic and “Antiarrhythmic” Phenotype

Cardiac atrophy is defined as a reduction in myocardial mass induced by various factors (see the previous chapter) that cause the wasting of cardiac muscle due to hemodynamic and/or metabolic stresses. Accelerated muscle protein degradation primarily occurs as a consequence of the activation of the two major proteolytic pathways, the muscle-specific ubiquitin–proteasomal pathway and the autophagic–lysosomal pathway, both contributing to the loss of muscle mass [91]. Myocardial atrophy is characterized by a decrease in the size and contractility of cardiomyocytes. Many molecular mechanisms of cardiac hypertrophy have been explored and established, but much less is known regarding the molecular mechanisms of cardiac atrophy. Thus, the pathogenesis of cardiac atrophy is still not completely understood and further studies are needed to the decrease knowledge gap.

Interestingly, the transcriptional signatures of atrophy and hypertrophy, including sarcomere proteins, ion pumps, and metabolic proteins, are the same [4], as shown in the foetal gene programme. Moreover, common microRNA signatures in cardiac hypertrophic and atrophic remodelling are induced by changes in hemodynamic load [92]. Additionally, adrenergic and angiotensin receptor activation are important stressors of the heart in both haemodynamic and metabolic load [4], as well as adrenergic and angiotensin receptor activation. Thus, the structural remodelling of the heart occurs in response to a decrease in its haemodynamic load (space flight, bed rest, heterotopic transplantation, LVAD) and metabolic load (starvation, malnutrition, cachexia) and/or an increase in metabolic and oxidative stress and low-grade inflammation (diabetes, obesity, altered thyroid status). Of note, the atrophic process may include cardiomyocytes, as well as vessels and capillaries [61,93] impacting myocardial perfusion.

The heart is capable of remodelling in response to workload by modulating protein synthesis and degradation [4]. Atrophic remodelling of the heart simultaneously activates pathways of protein synthesis and degradation [94,95,96]. Heterotopic cardiac transplantation decreases the capacity for myocardial protein synthesis [97]. Experimental spinal cord injury causes left ventricular atrophy associated with an upregulation of proteolytic pathways [98]. Accordingly, an increase in muscle ring finger-1 and Beclin-1 protein levels indicates an upregulation of the ubiquitin–proteasome system and autophagy–lysosomal machinery. The hallmark of cardiac atrophy is the increase in protein degradation [64] and activation of ubiquitin proteasome system (UPS)-mediated proteolysis [98,99]. Consequently, this activates autophagy for protein clearance in cardiomyocytes [91,100]. This process, generally considered to be cardio-protective, is augmented due to starvation [101] or cancer-associated cachexia and is more pronounced in males than females [102]. Autophagy is essential for cardiomyocyte homeostasis; however, chronic, excessive and dysregulated autophagy during stress not only promotes myocardial atrophy [100] but can also lead to cardiomyocyte death [101,103]. Indeed, matricellular proteins thrombospondins induced lethal cardiac atrophy through endoplasmic reticulum stress effector PERK-ATF4-regulated autophagy [104]. The inhibition of autophagy recovers cardiac dysfunction and unloading-induced atrophy [105]. Meanwhile, autophagy is essential for mediating the regression of hypertrophy during the unloading of the heart [106]. It can be hypothesized that autophagy, via its protein clearance, may affect structural remodelling and arrhythmia triggers and is thereby implicated in the susceptibility of the heart to arrhythmias, as recently suggested [107].

The specific over-expression of the ubiquitin ligase MAFbx/Atrogin-1 in the heart has been reported to inhibit the development of cardiac hypertrophy and is required for atrophic remodelling [108]. During unloading, MAFbx/Atrogin-1 represses calcineurin-induced cardiac hypertrophy. AFbx/Atrogin-1 not only regulates protein degradation, but also reduces protein synthesis, exerting a dual role in regulating cardiac mass. The increased expression of WW domain-containing E3 ubiquitin protein ligase 1 resulted in microgravity induced myocardial atrophy [109] that was alleviated by its deficiency. Striated muscle-specific ubiquitin ligase MuRF1 is important in regulation of cardiomyocyte size through alterations in protein turnover, and its loss is associated with doxorubicin- and dexamethasone-induced cardiac atrophy [82,110].

Leptin activated the anti-hypertrophic kinase GSK3β and increased the protein levels of muscle-specific ubiquitin ligases, MuRF1 and muscle atrophy in rats [111]. Ischemic stroke induced atrophy along with increased atrogin-1 and E3 ubiquitin ligase MuRF-1 and altered the transcriptome profile in mice, resulting in cardiac dysfunction [53]. On the other hand, MuRF1 was not required to induce cardiac atrophy in intact mice [110]. Considering the impact of ubiquitin ligases on ion channels function [112], it is challenging to elucidate the implication of MuRF1 in this process in atrophied heart. Moreover, ubiquitin ligases via the modulation of Ca^2+^ handling [113], which has pro-arrhythmic disorders that may impact arrhythmogenesis. 

Amino acid taurine depletion led to cardiomyopathy with myocardial atrophy [114]. Atrophic remodelling of the heart was associated with the increased expression of IGF-1 and FGF-2 and decreased nuclear receptor PPARα transcript levels along with PPARα-regulated genes. Consistent with the increase in IGF-1, transcript levels Mafbx/Atrogin-1 and MuRF-1 were decreased [95]. Furthermore, unloading was associated with the phosphorylation of ERK1, STAT3, and p70S6K.

Mechanical unloading of the failing human heart resulted in an increase in calpain 2 gene expression. Likewise, transcript levels of calpain-1 and -2, calpain activity and a calpain-specific degradation product were all increased in the unloaded rat heart [115]. Thus, the calpain proteolytic system is activated in mechanically unloaded heart. Moreover, cardiac sympathetic neurons are strong regulators of the cardiomyocyte size via the β2-adrenergic receptor-dependent repression of proteolysis [116]. This is of great clinical relevance given the role of β-adrenergic receptors in cardiovascular diseases and their modulation in therapy.

In addition, there is an association between proteolysis and oxidative stress and inflammation in the process of ‘cardiac cachexia’ [50,68], which may impact arrhythmogenesis. Tumour-derived cytokines, such as IL-6, TNFα and IL-1, and the activation of transcription factor NF-κB induce sarcomere proteolysis [117], while inhibiting NF-κB signalling largely prevents cancer-induced muscle wasting. This indicates its prominent role in NF-κB muscle atrophy.

Interestingly, TRPC3-Nox2 axis mediates nutritional deficiency-induced cardiomyocyte atrophy via the activation of extracellular ATP and proinflammatory purinergic signalling [62,118]. An inflammatory cardiac fibroblast phenotype underlies chronic alcohol-induced cardiac atrophy and dysfunction [51]. Notably, the activation of pro-inflammatory purinergic signalling is deleterious to the heart and may increase cardiac arrhythmia susceptibility [119]. Both oxidative stress and inflammation, along with the activation of purinergic signalling, are crucial factors involved in the development of cardiac arrhythmias [54].

On the other hand, NLRP3/IL-1β inflammasome pathway inhibition attenuates cardiac atrophy and cardiomyopathy in sepsis [52]. Moreover, the inhibition of Ca^2+^ handling due to hypocaloric feeding has been reported [63], which may prevent Ca^2+^-overload-induced arrhythmias [120]. On the other hand, trans-sarcolemmal Ca^2+^ fluxes were increased in the cardiac atrophy of the unloaded heterotopically transplanted rat heart [121,122]. Moreover, Ca^2+^ pump, Na^+^-Ca^2+^ exchange and Ca^2+^-transient activities of cardiomyocytes affecting cardiac function were increased in early cardiac atrophy, suggesting an adaptive mechanism to maintain contractile function [123]. It would be interesting to elucidate the implication of altered Ca^2+^ handling in atrophied myocardium in ventricular arrhythmia susceptibility. 

Myocardial atrophy is also characterized by the deficiency of myomesin-1, which is important in sarcomere assembly, contractility regulation and the development of cardiomyocytes [124]. The disorientation and loss of contractile filaments, as well as loss of Z-line substance, were observed [1]. Interestingly, cardiomyocyte sarcolemma from unloaded transplanted had higher contents of phosphatidic acid, sphingomyelin, and cholesterol [122]. The progressive depletion of titin (which acts as a mechanosensor) leads to sarcomere disassembly, atrophy, and eventually, dilated cardiomyopathy [125].

Moreover, cardiac atrophy after mechanical unloading exaggerates the proportion of total collagen that is responsible for diastolic dysfunction [105]. Indeed, acute mechanical unloading significantly increases hydroxyproline in connective tissue, indicating an early increase in connective tissue/muscle mass ratio [1]. Meanwhile, collagen deposition and fibrosis underlie a pro-arrhythmic substrate [54,126] that may increase the risk of develop malignant ventricular arrhythmias. However, despite altered Ca^2+^ handling and fibrosis due to chronic unloading, the incidence of malignant arrhythmias are rare in atrophied hearts with depressed contractility.

The activation of protein kinase C (PKC) through diacylglycerol is recognized as an early and common mechanism leading to cardiac dysfunction and remodelling in type-1 diabetes, while the suppression of this activation by diacylglycerol kinase in diabetic mice inhibits myocardial atrophy [33]. Notably, a specific increased expression of PKCε isoform in atrophied type-1 diabetic rat hearts is associated with upregulation and phosphorylation of electrical coupling protein Cx43 [18], thereby contributing to protection from VF. Likewise, an atrophied hypothyroid rat heart characterized by an increase in both PKCε and Cx43 was resistant to inducible VF [19]. Moreover, atrophied cardiomyocytes of diabetic or hypothyroid rat hearts did not exhibit apparent pro-arrhythmic alterations in topology (lateralization) of Cx43 [18], unlike hypertrophied cardiomyocytes, which exhibit pro-arrhythmic Cx43 redistribution from intercalated discs to the lateral plasma membrane [15,127,128]. Mechanical forces are most likely involved in the remodelling of cardiomyocyte Cx43 distribution [129]. Meanwhile, depressed mechanical forces in atrophied heart may prevent abnormal Cx43 topology and contribute to a decreased susceptibility to developing malignant cardiac arrhythmia. In contrast, the accelerated disruption of cardiac adiponectin-Cx43 signalling is a molecular mechanism for exacerbated cardiac dysfunction and pro-arrhythmias in type-2 diabetic females [130].

Non-coding RNA molecules are involved in doxorubicin-mediated cardiotoxicity, atrophy and heart failure [75], either via the upregulation of miR-208a, miR-532-3p or downregulation miR30e, and a Mhrt-myosin heavy-chain-associated RNA transcript. Moreover, RNA binding proteins, such as quaking, are downregulated due to doxorubicin, thereby promoting atrophy [131]. In contrast, the overexpression of quaking-5 strongly attenuated the cardiotoxicity of doxorubicin by regulating a set of circular RNAs.

## 5. Consequences of Myocardial Atrophy

Data in the literature suggest that there is an acute and chronic stage of myocardial atrophic remodelling, while both are accompanied by oxidative stress and inflammation deteriorating cardiodepression. Cardiac atrophy is the prominent change after mechanical unloading and heart dysfunction and aggravates overtime due to disorders in interstitial matrix homeostasis and fibrosis [132]. Although there is decreased myocardial volume and increased stiffness along with reduced cardiac output, contractile capacity is preserved even in the long-term unloaded heart [59]. However, excessive unloading-associated myocardial atrophy and fibrosis may adversely affect the process of reverse remodelling. In addition, atrophy affects capillaries and vessels and the insufficient perfusion of myocardial mass might facilitate heart failure [61,93].

Moreover, data in the literature suggest that, different from the hypertrophic heart, the atrophic heart is not vulnerable to life-threatening ventricular arrhythmias [15,17,18,19,20]. Experimental studies indicate that the upregulation of PKCε/Cx43 signalling is most likely a crucial mechanism supporting myocardial electrical stability. It appears that a reduced heart rate and depressed contractility, the upregulation of Cx43, and the maintenance of its topology in atrophied myocardium underlies the antiarrhythmic phenotype despite Ca^2+^ handling alterations. This issue requires further attention because cardiac arrhythmias remain a clinical challenge due to the alarming increase in atrial fibrillation and incidence of VF contributing to high morbidity and mortality in developed countries [21]. 

In this context, it should be noted that, in contrast to insulin-dependent type-1, diabetes mellitus resulting in myocardial atrophy, the insulin-resistant type-2 diabetes mellitus did not exhibit cardiac atrophy and is susceptible to both atrial and ventricular arrhythmias [133,134]. The increase in atrial volume and disordered calcium handling due to alterations in sarcoplasmic reticulum calcium ATPase (SERCA2) levels may be involved in the occurrence of atrial fibrillation [134]. Meanwhile, insulin secretion may be affected by the type 2 ryanodine receptor Ca^2+^ release channels [133]. These factors affecting calcium homeostasis seem to be involved in the arrhythmogenesis of a diabetic heart.

## 6. Prevention, Attenuation or Reversion of Myocardial Atrophy

Data in the literature strongly indicate that both hypertrophy and atrophy may be potentially reversible via modulation of mechanical/metabolic loading [4,25,36]. In fact, the contractile function of unloaded hearts is preserved, despite atrophic remodelling [59,135] suggesting a potential for “reverse remodelling” upon reloading. However, in the absence of compensatory increases in contractile function, reductions in myocardial mass will lead to an impaired overall work capacity and cardiodepression resulting in heart failure. Therefore, these approaches, as well as cellular and molecular signalling, that promote the reversion or attenuation of myocardial atrophy and decrease contractility, facilitating heart failure, would be of great interest. In general, smaller myocytes exhibit the highest growth potential, whereas larger myocytes exhibit the highest potential to atrophy [2]. Partial and total unloading affects the myocardial remodelling of non-failing hearts in a rodent model to different extents in myocardial atrophy, fibrosis, glucose metabolism and mechanical study [132] Thus, cardiac recovery is possible by means of mechanical loading [136,137] yet remains rare. In fact, therapeutic approaches that aim to prevent/attenuate myocardial atrophy along with associated abnormalities are still rare, and these abnormalities can result in depressed contractility. Therefore, more attention should be paid to this important issue.

Humoral factors, including thyroid hormone signalling may affect cardiomyocyte growth and contractility via the regulation of protein expression [17,36]. It would be of interest to explore whether any mechanical and/or metabolic unloading (including cases listed in Section 3 is accompanied by thyroid hormone deficiency. Meanwhile, patients suffering from heart failure due to pronounced myocardial structural remodelling exhibit a deficit in thyroid hormones or low tri-iodothyronine syndrome [138,139]. These individuals, as well as hypothyroid and type-1 diabetic patients, may benefit from the recovery of thyroid hormone homeostasis [140]. Hypothyroidism-induced cardiac atrophy is reversible [27], and a myocardial mass reduction in type-1 diabetic rats was attenuated by supplementation with thyroid hormones [18]. Moreover, the restoration of cardiac function, associated with the suppression of myocardial atrophy, was achieved by diacylglycerol kinase zeta via the modulation of PKC activity and insulin signalling in a diabetic rat model [33]. Interestingly, the upregulation of PKCε in the atrophic left ventricle of type-1 diabetic rats was supressed along with an increase in myocardial mass by treatment with thyroid hormones [18]. Reverse myocardial remodelling has also been reported in a rat model of aldosteronism-induced cachexia after complete hormone withdrawal [141]. Meanwhile, the administration of angiotensin II increased total cardiac protein synthesis in an adult denervated rat heart, leading to an attenuation in cardiac atrophy [142].

Cardiac contractility was enhanced by the phosphorylation of myosin light chain 2 (MLC2) by cardiac-specific MLC kinase, while its loss led to cardiac atrophy in vitro and in vivo [143]. Potential treatments of myocardial depression caused by a deficiency of myomesin-1, which is important in sarcomere assembly and contractility regulation [124], require further study. Likewise, transient receptor potential channels, namely the TRPC3-Nox2-based protein signalling complex have recently been discussed as therapeutic targets of myocardial atrophy and heart failure [118]. The reverse remodelling of small dedifferentiated cardiomyocytes bordering fibrosis was achieved with assisted recovery [144] suggesting potential for functional myocardial regeneration. On the other hand, ventricular pacing attenuated but did not reverse cardiac atrophy or isomyosin shift in the heterotopically transplanted rat heart [145]. Cardiac atrophy and fibrosis associated with the upregulation of miR-21 in a heart failure rat model were ameliorated after injection of miR-21 antagonist [146].

D-3-hydroxybutyrate, one of the main components of the ketone with a potent anti-catabolic function, has clear hemodynamic effects in atrophic cardiomyocytes and exerts beneficial metabolic reprogramming effects. The anabolism/catabolism balance of muscle protein was maintained with D-3-hydroxybutyrate, via the Akt/FoxO3a and mTOR/4E-BP1 pathways [147]. Exogenous D-3-hydroxybutyrate exerts clear hemodynamic effects on patients with chronic heart failure [148,149].

Considering autophagy as an important process for maintaining cardiac homeostasis and survival mechanism [101,103], the suppression/regulation of excessive autophagy in an unloaded heart may be a therapeutic target to prevent, attenuate or reverse cardiac atrophy. Accumulating evidence suggests that dysregulated autophagy is associated with heart failure. The inhibition of autophagy recovered cardiac dysfunction and attenuated atrophy in rat heart tissue [105].

The total mechanical unloading of the heart models of heterotopic heart transplantation leads to cardiac atrophy and functional deterioration. Meanwhile, the partial unloading of animal or human failing hearts with left ventricular assist devices can ameliorate heart failure symptoms in some patients [132,150]. Combined clenbuterol (β2- AR agonist) and metoprolol (β1-AR blocker) therapy showed superior functional effects to mono-therapy in reverse remodelling during mechanical unloading in the rodent model [151].

Notably, emerging evidence indicates that the Hippo pathway, which controls cell proliferation, apoptosis and differentiation, is critical in cardiac homeostasis, disease, and regeneration. Targeting the Hippo pathway has tremendous potential as a therapeutic strategy for treating intractable cardiovascular diseases, such as heart failure, as recently reviewed [152].

Preventing or reversing cardiac atrophy may mitigate the cardiac dysfunction of cancer patients receiving anthracyclines [82]. A novel strategy may be to inhibit doxorubicin-induced atrophy and promote cardiomyocyte hypertrophy via the modulation of microRNAs [75]. Notably, the novel in vivo potential of calmoduline antagonist, trifluoperazine, which ameliorates doxorubicin-induced structural remodelling and cardiodepression through the suppression of NF-κB and apoptosis, has been reported [153]. Non-invasive transcutaneous vagal nerve stimulation downregulates pro-inflammatory chemokine-related genes and prevents pro-inflammatory macrophages recruitment in the cardiac tissue of rats exposed to doxorubicin and improved myocardial performance [80]. Likewise, NLRP3/IL-1β inflammasome pathway inhibition attenuates cardiac atrophy in sepsis [52]. Moreover, there was a benefit of treatment with flavonoid resveratrol via the inhibition of transcription factor NF-κB [117]. The inhibition of cyclooxygenase-2, ubiquitous central proinflammatory mediator, attenuated neurohumoral activation in anthracyline-induced atrophy and heart failure [154]. The administration of oxidative stress targeting flavonoid quercetin, did not affect myocardial atrophy but attenuated the adverse effects of doxorubicin, including cardiomyocyte injury and apoptosis. It also influenced myocardial responses to acute ischemic stress in rat hearts [79]. Moreover, ethoxyquin, a competent radical-trapping antioxidant for preventing ferroptosis, ameliorated contractile dysfunction and myocardial atrophy in a murine model of doxorubicin cardiotoxicity [155]. Ibudilast, a drug approved for clinical use, attenuated doxorubicin-induced cytotoxicity by suppressing the formation of the TRPC3 channel and NADPH oxidase 2 protein complexes [156]. The pharmacological prevention of the TRPC3 -Nox2 protein complex can maintain cardiac flexibility in mice after anti-cancer drug treatment [118]. Treatment with enalapril preserved left ventricular morphology and function in a clinically relevant model of chronic doxorubicin-induced cardiotoxicity via increased stimulation of the PI3K/AKT/mTOR axis and normal connective tissue growth factor levels [157]. This suggests potential therapeutic implications. Long-acting thioredoxin ameliorated doxorubicin-induced cardiomyopathy in mice via its anti-oxidative and anti-inflammatory action [158]. Notably, even exercise training performed during doxorubicin-based chemotherapy in mice can be a valuable approach to attenuate cardiotoxicity, including the alleviation of myocardial atrophy [159]. The reversal of cancer cachexia and muscle wasting by ActRIIB antagonism leads to prolonged survival [160] implying that the ActRIIB pathway limits muscle growth.

## 7. Conclusions

Mechanical or metabolic unloading due to various conditions listed in this review activates atrophic signalling leading to a decrease in cardiomyocyte size and left ventricular mass associated with cardiodepression. Atrophic phenotype accompanied by oxidative stress and inflammation aggravates heart function over time due to extracellular matrix remodelling that may progress to heart failure. It is interesting that atrophic heart is less vulnerable to malignant ventricular arrhythmias. Available data suggest that reduced heart rate, depressed contractility, enhanced cardioprotective PKCε signalling and, particularly, the upregulation of electrical coupling protein Cx43, along with the maintenance of its topology, may underlie the anti-arrhythmic phenotype. This issue is challenging to elucidate in more detail, and it is perhaps difficult to discover a new strategy for malignant arrhythmia prevention or treatment. Moreover, data in the literature suggest that the prevention of heart failure via the attenuation or reversion of myocardial atrophy is possible, although this requires further research.

## Figures and Tables

**Table 1 biomedicines-10-02819-t001:** Conditions leading to cardiac atrophy and incidence of cardiac arrhythmias.

Unloading	Causes	Consequences	
		Mechanical Dysfunction	Incidence of Malignant Ventricular Arrhythmias
**Metabolic** **unloading**	Caloric restriction	+	No information
	Cancer cachexia	+	No information
	Doxorubicin treatment	+	Increase in risk of arrhythmias [90] *
	Sepsis	+	May increase risk of arrhythmias [52] *
	Chronic alcohol exposure	+	No information
**Haemodynamic** **unloading**	Prolonged mechanical support via left ventricular assist device	+	No information
	Spinal cord injury	+	May increase risk of arrhythmias [47] *
	Physical inactivity/ prolonged bed rest	+	No information
	Weightlessness/microgravity	+	No information
	Type 1 diabetes mellitus	+	No arrhythmias [18,20]
	Thyroid hormones deficiency	+	No arrhythmias [17,19]
	Heterotopic cardiac transplantation	+	No information

* In spinal cord injury cardiac arrhythmias are mostly attributed to the autonomic instability. Meanwhile, in sepsis, arrhythmias might be attributed to the inflammasome, which potentially deteriorates myocardial Cx43 levels. Doxorubicin treatment prolongs QT interval and the duration of the QRS complex due to ion channels disorders, deteriorating Ca^2+^ homeostasis and reducing Cx43 expression.

## Data Availability

Not applicable.

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
