# Peer review of "Does Myocardial Atrophy Represent Anti-Arrhythmic Phenotype?"

_biomedicines, 2022, doi:10.3390/biomedicines10112819_

Round 1

Reviewer 1 Report

This is a review paper discussing role of myocardial atrophy in reduced propensity to cardiac arrhythmias. The paper is well written, with up to date information, informative and easy to follow, though a bit chaotic. I have some comments:

1. The authors should discuss the role of physiological vs. pathological hypertrophy and atrophy (e.g., hypertension vs. endurance training/pregnancy and hypothyroidism/cancer cachexia vs. deconditioning). Do arrhythmias differ depending on physiological vs. pathological etiology?

2. The paper almost does not discuss arrhythmias at all. The authors should briefly discuss types and mechanisms of arrhythmias and potential effects of atrophy on those cellular mechanisms and arrhythmias itself both in the experimental and clinical setting.

Author Response

Responses to Reviewer 1

Thank you very much for your time to read our manuscript and we appreciate very much your comments and suggestions. Our responses are highlighted by red color in revised version. We believe that thanks to your impact the topic of the paper is more clear for readers and we hope that myocardial atrophy along with reduced propensity to cardiac arrhythmias may attract attention of cardiologists.

This is a review paper discussing role of myocardial atrophy in reduced propensity to cardiac arrhythmias. The paper is well written, with up to date information, informative and easy to follow, though a bit chaotic. I have some comments:

  1. The authors should discuss the role of physiological vs. pathological hypertrophy and atrophy (e.g., hypertension vs. endurance training/pregnancy and hypothyroidism/cancer cachexia vs. deconditioning). Do arrhythmias differ depending on physiological vs. pathological etiology?

Thanks to your comments we thought a bit more about the subject. We added 2 paragraphs and several references in Chapter 1, to discuss “physiological” and “pathological” myocardial hypertrophy, while considering myocardial atrophy as an adaptation of the heart to non-physiological conditions.

As for enlargement of the heart, we think that the term “hypertrophy” is used in case when adult cardiomocytes hypertrophy independent on whether due to sport activity, humoral factors or systemic disease. While myocardial hypertrophy itself renders the heart less electrically stable mostly due to connexin43 alterations, ion channels disorders and altered Ca2+ handling, independent on etiology. 

  1. The paper almost does not discuss arrhythmias at all. The authors should briefly discuss types and mechanisms of arrhythmias and potential effects of atrophy on those cellular mechanisms and arrhythmias itself both in the experimental and clinical setting.

We added new chapter 2 that describes briefly mechanisms and factors implicated in development of malignant cardiac arrhythmias (valuable for both experimental and clinical setting). We also outlined, in chapter 4 (previously chapter 3), some possible mechanisms underlying antiarrhythmic properties of atrophied heart. We did effort to include all relevant literature concern of myocardial atrophy but we realize that it is too little to understand deeply “antiarrhythmic phenotype”. Therefore, this issue is open for further research. We believe that it could provide new ideas how to prevent or fight life-threatening ventricular arrhythmias. 

Reviewer 2 Report

The authors summarized the mechanism of atrophy. This point of view is important to investigate the cause of arrhythmia. However, the reviewer has some concerns about the manuscript as bellow. Please consider and reply about them.

1. The reviewer strongly agrees with “While long-term myocardial atrophy accompanied by oxidative stress and inflammation along with extracellular matrix fibrosis leads to cardiac dysfunction, heart failure progression and perhaps transient ventricular arrhythmias” in line 118-120. Then, where does ischemic cardiac injury locate in Table 1? It is considered as the main cause of ventricular arrhythmia.

2. As authors described, Cx43 is crucial factor for arrhythmia, and it is much investigated in atrial fibrillation. According to the manuscript, Cx43 is important for the cause of diabetes, hypothyroid, and mechanical forces. Do authors have any information about doxorubicin treatment, sepsis, and spinal cord injury which may increase risk of arrhythmias in Table 1?

Author Response

Reviewer 2

We would like to thank you very much for your time to read our manuscript and for your comments and suggestions.

The authors summarized the mechanism of atrophy. This point of view is important to investigate the cause of arrhythmia. However, the reviewer has some concerns about the manuscript as bellow. Please consider and reply about them.

  1. The reviewer strongly agrees with “While long-term myocardial atrophy accompanied by oxidative stress and inflammation along with extracellular matrix fibrosis leads to cardiac dysfunction, heart failure progression and perhaps transient ventricular arrhythmias” in line 118-120. Then, where does ischemic cardiac injury locate in Table 1? It is considered as the main cause of ventricular arrhythmia.

Yes, we agree with you fully that myocardial ischemic injury is highly pro-arrhythmic, in particular in cases of myocardial infarction, which results in severe electrical disorders. However, in the context of myocardial atrophy we did not find in literature information about ischemic injury. According to our own experience, atrophied myocardial tissue of hypothyroid rats or type1 diabetic rats did not show severe ischemic injury rather moderate subcellular alterations of the cardiomyocytes. Therefore, we included in the text: The ischemic component impacting process of cardiomyocyte atrophy is not elucidated. 

  1. As authors described, Cx43 is crucial factor for arrhythmia, and it is much investigated in atrial fibrillation. According to the manuscript, Cx43 is important for the cause of diabetes, hypothyroid, and mechanical forces. Do authors have any information about doxorubicin treatment, sepsis, and spinal cord injury which may increase risk of arrhythmias in Table 1?

We did not find in literature data about implication of myocardial Cx43 in spinal cord injury, sepsis.  In spinal cord injury cardiac arrhythmias are mostly attributed to the autonomic instability. While in sepsis arrhythmias might be attributed to the inflammasome that potentially deteriorates myocardial Cx43 levels. Doxorubicin treatment prolongs QT interval, duration of QRS complex due to ion channels disorders and deteriorate Ca2+ homeostasis and reduced Cx43 expression. We included these remarks under the Table.

Round 2

Reviewer 1 Report

The authors have significantly impropved their manuscript. 

Author Response

Dear authors,
About the manuscript entitled “Does myocardial atrophy represent anti-arrhythmic phenotype?”, I would suggest to fix the following points:

Dear Academic Editor,

Thank you for your time to read our review manuscript. We appreciate your effort to increase the impact of the manuscript by adding several references. We have read the suggested papers but we did not find any association with myocardial remodeling or ventricular atrophy. The latter is the main issue we discussed in our manuscript, therefore, we pointed out the conditions resulting to this myocardial atrophy, including insulin-dependent type-1 diabetes mellitus unlike non-insulin dependent type-2 DM. The latter did not exhibit myocardial or ventricular atrophy and mostly is accompanied by other systemic diseases like obesity, dyslipidemia, hypertension.

We agree with you that myocardial alterations due to insulin resistant type-2 DM differ from insulin-dependent type-1 DM and the heart suffering from type-2 DM is not atrophic and more susceptible to both ventricular and atrial arrhythmias. Therefore, we added references Sardu et al. 2020 and Santuli et al. 2015 as an example of arrhythmogenesis in diabetic heart. We also included the reference Sardu et al. 2015 dealing with epigenetic microRNAs regulation of atrial fibrillation. Moreover, we added reference Leffler at al. 2017 pointing out disruption of adiponectin-connexin43 signaling that underlies exacerbated myocardial dysfunction in type-2 diabetic rats and likely contribute to pro-arrhythmia.

We hope you understand our effort to keep focus on myocardial atrophy rather than diabetic heart disease. While it appears that focused issue solely on diabetes mellitus should include all points you suggested. This is the reason why we did not follow all your suggestions. We are very sorry for that.

Please fully describe the-implication of hyperglycemia and insulin resistance in the regulation of calcium channels currents, and adverse electrical remodelling in diabetic patients, as main determinant of atrial (Modulation of SERCA in Patients with Persistent Atrial Fibrillation Treated by Epicardial Thoracoscopic Ablation: The CAMAF Study. J Clin Med. 2020 Feb 17;9(2):544. doi: 10.3390/jcm9020544), and ventricular arrhythmias (Calcium release channel RyR2 regulates insulin release and glucose homeostasis. J Clin Invest. 2015 May;125(5):1968-78. doi: 10.1172/JCI79273). In my opinion these points are relevant because they could explain the negative implication of hyperglycemia and insulin resistance in the regulation of channels, favouring electrical remodelling of cardiac structures and arrhythmogenicity.
Please describe this point and the suggested references.

= We added suggested references in the text highlighted with blue color.

-It is also relevant to report data about the implication of epigenetic in the regulation of atrial fibrillation ad in the post-ablative prognosis (microRNA expression changes after atrial fibrillation catheter ablation. Pharmacogenomics. 2015 Nov;16(16):1863-77. doi: 10.2217/pgs.15.117). please discuss this point. What is your opinion?

= We agree with you that microRNAs are important epigenetic factors that may be involved in susceptibility of the heart to arrhythmias and we revised the text according it including suggested reference.

-Again, a central role is played by the autonomic dysfunction to induce arrhythmias in patients with diabetes mellitus (Autonomic dysfunction is associated with brief episodes of atrial fibrillation in type 2 diabetes. J Diabetes Complications. 2015 Jan-Feb;29(1):88-92. doi: 10.1016/j.jdiacomp.2014.09.002), and in those with metabolic syndrome (Cardiac electrophysiological alterations and clinical response in cardiac resynchronization therapy with a defibrillator treated patients affected by metabolic syndrome. Medicine (Baltimore). 2017 Apr;96(14):e6558. doi: 10.1097/MD.0000000000006558). Please discuss this point.

= Because current review article is focused on myocardial atrophy that is the case of type-1 while not type-2 diabetes mellitus or metabolic syndrome, therefore, we would like to keep this main point avoiding further information.  

- Finally, do not forget the opportunity to find best treatments to control the over-inflammation/ oxidation, cardiac remodelling, and ameliorate clinical outcomes in T2DM with a por-arrhythmic status (Metabolic syndrome is associated with a poor outcome in patients affected by outflow tract premature ventricular contractions treated by catheter ablation. BMC Cardiovasc Disord. 2014 Dec 6;14:176. doi: 10.1186/1471-2261-14-176;
Multipolar pacing by cardiac resynchronization therapy with a defibrillators treatment in type 2 diabetes mellitus failing heart patients: impact on responders rate, and clinical outcomes.        Cardiovasc Diabetol. 2017 Jun 9;16(1):75. doi: 10.1186/s12933-017-0554-2). Please discuss it.

= Because current review article is focused on myocardial atrophy that is only a case of type-1 while not type-2 diabetes mellitus, therefore, we would like to keep this main point of this review article. Undoubtedly, the diabetic disease requires to pay more attention perhaps in focused review.     

-Improve English form of the text.

= We edited English.
